# Effects of Mineralocorticoid Receptor Blockade and Statins on Kidney Injury Marker 1 (KIM-1) in Female Rats Receiving L-NAME and Angiotensin II

**DOI:** 10.3390/ijms24076500

**Published:** 2023-03-30

**Authors:** Jiayan Huang, Ezgi Caliskan Guzelce, Shadi K. Gholami, Kara L. Gawelek, Richard N. Mitchell, Luminita H. Pojoga, Jose R. Romero, Gordon H. Williams, Gail K. Adler

**Affiliations:** 1Division of Endocrinology, Diabetes, and Hypertension, Department of Medicine, Brigham and Women’s Hospital, Harvard Medical School, Boston, MA 02115, USA; 2Department of Pathology, Brigham and Women’s Hospital, Harvard Medical School, Boston, MA 02215, USA

**Keywords:** N-omega-nitro-L-arginine methyl ester (L-NAME), angiotensin II (ANG II), mineralocorticoid receptor (MR), statin, simvastatin, pravastatin, kidney injury molecule (KIM-1), CVD (cardiovascular disease)

## Abstract

Kidney injury molecule-1 (KIM-1) is a biomarker of renal injury and a predictor of cardiovascular disease. Aldosterone, via activation of the mineralocorticoid receptor, is linked to cardiac and renal injury. However, the impact of mineralocorticoid receptor activation and blockade on KIM-1 is uncertain. We investigated whether renal KIM-1 is increased in a cardiorenal injury model induced by L-NAME/ANG II, and whether mineralocorticoid receptor blockade prevents the increase in KIM-1. Since statin use is associated with lower aldosterone, we also investigated whether administering eiSther a lipophilic statin (simvastatin) or a hydrophilic statin (pravastatin) prevents the increase in renal KIM-1. Female Wistar rats (8–10 week old), consuming a high salt diet (1.6% Na+), were randomized to the following conditions for 14 days: control; L-NAME (0.2 mg/mL in drinking water)/ANG II (225 ug/kg/day on days 12–14); L-NAME/ANG II + eplerenone (100 mg/kg/day p.o.); L-NAME/ANG II + pravastatin (20 mg/kg/day p.o.); L-NAME/ANG II + simvastatin (20 mg/kg/day p.o.). Groups treated with L-NAME/ANG II had significantly higher blood pressure, plasma and urine aldosterone, cardiac injury/stroke composite score, and renal KIM-1 than the control group. Both eplerenone and simvastatin reduced 24-h urinary KIM-1 (*p* = 0.0046, *p* = 0.031, respectively) and renal KIM-1 immunostaining (*p* = 0.004, *p* = 0.037, respectively). Eplerenone also reduced renal KIM-1 mRNA expression (*p* = 0.012) and cardiac injury/stroke composite score (*p* = 0.04). Pravastatin did not affect these damage markers. The 24-h urinary KIM-1, renal KIM-1 immunostaining, and renal KIM-1 mRNA expression correlated with cardiac injury/stroke composite score (*p* < 0.0001, Spearman ranked correlation = 0.69, 0.66, 0.59, respectively). In conclusion, L-NAME/ANG II increases renal KIM-1 and both eplerenone and simvastatin blunt this increase in renal KIM-1.

## 1. Introduction

KIM-1, a type I transmembrane glycoprotein, is expressed in low levels in the proximal tubules of the healthy kidney [1]. Kidney injury leads to dedifferentiation of the epithelium and increased expression of KIM-1 mRNA and protein. KIM-1 accumulates on the apical membrane of injured proximal tubules where it is involved in regeneration and repair [1,2,3,4]. Urinary and plasma KIM-1 are markers of renal proximal tubule injury in humans and rodents [5,6,7,8,9,10,11,12]. KIM-1 increases in ischemia-reperfusion models and after exposure to toxic compounds, e.g., adriamycin, cisplatin, cyclosporin, gentamicin [1,8,9,10,13,14]. KIM-1 is proposed as a more specific marker for renal tubular injury than traditional renal markers—serum creatinine, serum urea nitrogen and creatinine clearance [10]—and it has been approved by the United States Food and Drug Administration and the European Medicines Agency for preclinical assessment of nephrotoxicity [15]. In humans, its utility as a biomarker has been demonstrated in acute [16] and chronic renal injury, as an excellent predictor of renal disease progression [5,17]. KIM-1 is also shown to be a strong predictor of cardiac injury [18,19,20,21,22,23,24,25].

In a study of hospitalized patients with acute renal failure, higher urinary KIM-1 was associated with a higher risk of dialysis requirement or death [26] and higher urinary levels of KIM-1 provided prognostic information regarding mortality risk in patients with heart failure [27]. In addition, higher urinary KIM-1 was associated with an increased risk for cardiovascular mortality independent of established cardiovascular risk factors, eGFR, and albuminuria in a community-based Uppsala Longitudinal study of adult men with a mean age 77 years [20]. Plasma KIM-1 was also shown to predict the future decline of eGFR [28,29,30]. Further, plasma KIM-1 was strongly associated with the presence of anatomically significant CAD in the CASABLANCA dataset [23].

Albuminuria leads to proximal tubule injury; however, selective proximal tubule injury can develop independent of albuminuria and then lead to albuminuria [31]. It is established that aldosterone increases albuminuria and blockade of aldosterone’s receptor, the mineralocorticoid receptor (MR), reduces albuminuria [32,33]. Further, MR blockade improves renal outcomes in large-scale clinical trials [34,35]. However, the impact of MR activation and MR blockade on proximal tubule injury is less certain. In preclinical studies, administration of angiotensin II (ANG II), which stimulates aldosterone, increases KIM-1 [36,37]; however, these studies did not assess the impact of MR blockade on KIM-1. Renal KIM-1 mRNA levels were increased in the hypertensive DOCA-salt model, but treatment with the MR blockade, spironolactone did not significantly prevent the increase in KIM-1, raising the possibility that the increases in KIM-1 were not due to MR activation but due to other factors such as increased blood pressure [38].

Therefore, the goal of this research is to determine the impact of modulating MR activity on proximal tubular injury as assessed by changes in KIM-1 levels by using the high salt diet, L-NAME/ANG II rodent model [39,40,41,42]. This model induces substantial cardiac and renal damage via mechanisms similar to that observed in heart failure, resistant hypertension, and salt sensitive hypertension. Yet, with inhibition of the MR or adrenalectomy, the injury is substantially prevented, even though blood pressure and aldosterone levels are minimally, or not at all, modified [39,40,41,42]. KIM-1 was assessed by three methods: KIM-1 shedding in 24-h urine collections, renal cortical KIM-1 mRNA, and renal KIM-1 immunohistochemistry. We used two experimental approaches to reduce MR activity. First, we administered the selective MR blocker, eplerenone, as has been used previously [41,42]. Second, we administrated simvastatin. The rationale for using simvastatin is that we previously demonstrated that the lipophilic statin, simvastatin, but not the hydrophilic statin, pravastatin, inhibits aldosterone production from isolated rat adrenal glomerulosa cells and that, in humans, simvastatin, but not pravastatin, use is associated with lower aldosterone levels [43]. Our experimental paradigm is shown in Figure 1.

## 2. Results

Three animals in the L-NAME/ANG II group and one animal in the L-NAME/ANG II/pravastatin group showed evidence of stroke, inability to move one side of the body after two to three days of ANG II treatment and were promptly euthanized. Blood and tissue were collected (N = 2, on day 14 at approximately noon; N = 2, on day 15 at 8 AM). All other animals were euthanized with collection of blood and tissue after three days of ANG II (N = 46, on day 15 between 8.30 and 11 am).

### 2.1. Blood Pressure 

Baseline systolic blood pressure was similar in all groups (*p* = 0.9, by one-way ANOVA, shown on Figure 2). Within each group, blood pressure obtained on day 6 and day 11 were similar (by a mixed model for repeated measures, *p* = 0.5 for repeated measures). Therefore, we averaged the blood pressure measurements on day 6 and day 11 to assess the effect of L-NAME on blood pressure. In general, the average systolic blood pressure on day 6 and day 11 was higher in groups receiving L-NAME compared to the control group, except for the group receiving L-NAME along with eplerenone (*p* = 0.002, by one-way ANOVA, and pairwise comparisons shown on Figure 2).

Systolic blood pressure assessed on day 14 was increased compared to baseline blood pressure in all L-NAME/ANG II/groups (L-NAME/ANG II group: *p* = 0.04, L-NAME/ANG II//eplerenone: *p* < 0.001, L-NAME/ANG II/pravastatin: *p* = 0.002, L-NAME/ANG II/simvastatin: *p* < 0.001 by paired student t-test), but not in the control group (*p* = 0.87, by paired Student’s t-test) (Figure 2). Analysis of day 14 systolic blood pressure by one-way ANOVA indicated significant differences between groups (*p* < 0.0001), and the comparisons for all pairs using the Tukey–Kramer HSD showed no significant differences in blood pressure between the L-NAME/ANG II treatment groups. However, all L-NAME/ANG II treatment groups had significantly higher blood pressure on day 14 as compared with the control group except for the L-NAME/ANG II group where the *p* value was 0.07 (Figure 2). The marginal significance in this latter group is likely due to having day 14 blood pressure in only eight animals as two animals died prematurely.

Body weights prior to treatment with ANG II were similar between the five groups (Table 1) but decreased with initiation of ANG II in all L-NAME/ANG II groups. We analyzed day 15 heart and kidney weight with and without correction for average body weight assessed on days 11 and 12 prior to ANG II infusion. Heart and kidney weights were similar between L-NAME/ANG II groups (Table 1).

### 2.2. Hormone Measurements

Plasma aldosterone levels assessed on day 15 were increased in all L-NAME/ANG II groups as compared with the control group (Table 2). PRA levels were lower in all L-NAME/ANG II groups compared to the control group, except in the eplerenone group (Table 2). The latter observation is consistent with the known effect of mineralocorticoid blockade to activate the renin-angiotensin-aldosterone system. On day 11, prior to initiation of ANG II, 24-h urinary aldosterone was only increased in the eplerenone group as compared with the control group, and not in the groups treated with L-NAME. The average 24-h urinary aldosterone during the three-days of ANG II treatment, and the day 15 adrenal cortex CYP11B2 mRNA levels were significantly increased in all L-NAME/ANG II groups as compared with the control group (Table 2). Treatment with eplerenone, simvastatin or pravastatin along with L-NAME/ANG II did not significantly affect plasma aldosterone, 24-h urinary aldosterone or CYP11B2 mRNA levels as compared with the L-NAME/ANG II group. However, the L-NAME/ANG II/eplerenone group had significantly higher average 24-h urinary aldosterone during the three days of ANG II treatment than L-NAME/ANG II/simvastatin (Table 2).

Plasma corticosterone levels were increased in only two of the L-NAME/ANG II groups, L-NAME/ANG II and L-NAME/ANG II/pravastatin groups, as compared to the control group. Further, the L-NAME/ANG II/eplerenone and L-NAME/ANG II/simvastatin groups had significantly lower corticosterone levels as compared to the L-NAME/ANG II group (Table 2).

### 2.3. 24-h Urinary KIM-1 and Albumin Measurements

The 24-h urinary KIM-1 was determined on day 11, the day before initiation of ANG II treatment, and day 14, during the third day of ANG II treatment. The 24-h urine collected on day 11 had a similar KIM-1/creatinine ratio (ng/mg) across all five groups (*p* = 0.08). With the administration of ANG II, there was a significant increase in day 14 urinary KIM-1/creatinine ratio (ng/mg) in all L-NAME/ANG II groups as compared to the control group (Figure 3). Pravastatin treatment did not affect day 14 urinary KIM-1/creatinine. In contrast, day 14 urinary KIM-1/creatinine ratio (ng/mg) was significantly lower in L-NAME/ANG II groups receiving eplerenone (*p* = 0.0046) and simvastatin (*p* = 0.031) as compared with the L-NAME/ANG II group.

Urine albumin/creatinine ratio on day 14 was significantly increased in all L-NAME/ANG II groups as compared to the control group, with no significant differences between the L-NAME/ANG II groups (Table 2).

### 2.4. KIM-1 Immunostaining of Kidney

KIM-1 immunostaining of kidney demonstrated minimal to no KIM-1 expression in the control group. In the L-NAME/ANG II group, KIM-1 immunostaining was present in proximal renal tubules (Figure 4).

Semi-quantitative analysis of KIM-1 immunostaining revealed significant increases in KIM-1 in all L-NAME/ANG II treated groups as compared with the control group (Figure 5). KIM-1 immunostaining was similar in the L-NAME/ANG II and L-NAME/ANG II/pravastatin groups. In contrast, KIM-1 immunostaining was significantly lower in L-NAME/ANG II groups receiving eplerenone (*p* = 0.004) and simvastatin (*p* = 0.037) as compared with the L-NAME/ANG II alone (Figure 5).

### 2.5. KIM-1 mRNA Expression of Kidney

All L-NAME/ANG II groups had higher KIM-1 mRNA expression in renal cortex as compared with the control group (Figure 6). KIM-1 mRNA expression was significantly lower in L-NAME/ANG II group receiving eplerenone (*p* = 0.012), as compared with the L-NAME/ANG II group (Figure 6).

### 2.6. Cardiac Injury/Stroke Composite Score

#### Cardiac Injury Histopathological Examination

Histological evaluation of the H&E-stained cardiac ventricles from L-NAME/ANG II treated animals revealed cardiac damage characterized by myocyte eosinophilia, nuclear drop-out, myocyte necrosis, and inflammatory infiltrates (Figure 7). Using a semi-quantitative score taking into account the extent of cardiac injury and stroke status, we demonstrated that all L-NAME/ANG II groups had significantly higher cardiac injury/stroke composite scores as compared to the control group (Figure 8). The cardiac injury/stroke composite score was similar in the L-NAME/ANG II/simvastatin, L-NAME/ANG II/pravastatin, and L-NAME/ANG II groups. In contrast, the L-NAME/ANG II/eplerenone group had a significantly lower cardiac injury/stroke composite score compared with the L-NAME/ANG II group.

Urinary KIM-1 correlated with the cardiac injury/stroke composite score across all groups (*p* < 0.0001, Spearman ranked correlation = 0.69) as well as within L-NAME/ANG II groups (*p* = 0.004, Spearman ranked correlation = 0.44). Similarly, renal KIM-1 immunostaining, and renal KIM-1 mRNA expression correlated with the cardiac injury/stroke composite score across all groups (*p* < 0.0001, Spearman ranked correlation = 0.66; *p* < 0.0001, Spearman ranked correlation = 0.59, respectively) and within L-NAME/ANG II groups (*p* = 0.01, Spearman ranked correlation = 0.40; *p* = 0.02, Spearman ranked correlation = 0.38, respectively).

## 3. Discussion

The L-NAME/ANG II model of cardiorenal damage was studied in female rats. L-NAME/ANG II significantly increased systolic blood pressure, aldosterone, cardiac injury/stroke composite score, albuminuria and renal proximal tubular damage characterized by increased renal KIM-1 immunostaining, urinary KIM-1 and renal KIM-1 mRNA expression. Treatment with eplerenone significantly reduced L-NAME/ANG II induced renal injury (i.e., reduced KIM-1) as well as the cardiac injury/stroke composite score. These data suggest that activation of MR causes proximal tubule damage that can be prevented with blocking the mineralocorticoid receptor. Further, treatment with simvastatin, but not pravastatin, significantly reduced renal KIM-1 immunostaining and urinary KIM-1 in animals receiving L-NAME/ANG II, suggesting that simvastatin protects against proximal tubular injury. Consistent with previous reports, the increase in blood pressure with administration of both L-NAME and ANGII was not modified by any treatments, supporting the hypothesis that in this model the protective effects were not mediated via a reduction in blood pressure. Further, the beneficial effects of simvastatin did not appear to be mediated by a measurable decrease in aldosterone. Assuming stress is associated with cardiovascular and renal damage, the plasma corticosterone levels support the beneficial effects of eplerenone and simvastatin. Corticosterone levels were similar to control in the eplerenone and simvastatin groups in contrast to the two limbs where damage was not reduced.

KIM-1 is expressed in proximal tubular cells in response to a variety of insults, including renal ischemia/reperfusion, adriamycin, cyclosporin, and diabetes. It is a biomarker of acute kidney injury as well as a predictor of chronic kidney injury and cardiovascular disease [16,17,18,19,20,21,22,23,24,25]. In the current study, we demonstrated that, in the setting of inhibition of NO production, infusion of ANG II increases aldosterone and increases protein expression of KIM-1 in proximal tubules. The proximal tubule KIM-1 expression is reduced by treatment with MR blockade indicating that in this model activation of the MR promotes proximal tubular damage. A strength of our study is that we assessed three measures of KIM-1: urinary KIM-1, renal KIM-1 immunostaining, and renal cortical KIM-1 mRNA expression. All of these KIM-1 measures were increased with L-NAME/ANG II and the increases were prevented with MR blockade. This result is consistent with the studies showing a decrease in KIM-1 with MR blockade in rodent models of renal injury-ischemia/reperfusion and diabetes mellitus and an increase in KIM-1 in the DOCA salt model [44,45]. Together these studies suggest that MR blockade reduces tubulointerstitial damage. Inability of MR blockade to reduce KIM-1 in the DOCA salt model is likely related to inadequate MR blockade relative to the amount of DOCA or to sample size [38].

The current study was performed in a model of aldosterone mediated cardiac and renal damage, the high salt, L-NAME/ANG II model. This is a model in which administration of L-NAME and ANG II in the presence of high salt diet raises blood pressure and causes cardiovascular and renal damage. ANG II is a known aldosterone secretagogue, and we have shown previously that the administration of a MR blockade (which blocks the action of aldosterone) significantly reduces cardiovascular and renal damage in the L-NAME/ANG II model but seldom reduces blood pressure [39,40,41,42]. Consistent with these findings, in the current study, the benefit of MR blockade on KIM-1 occurred without a decrease in blood pressure suggesting that the effects of MR blockade on proximal tubules are not mediated by blood pressure changes. Further MR blockade reduced KIM-1 without reducing albuminuria, suggesting a dissociation between glomerular damage and proximal tubular damage. This result is consistent with data indicating that proximal tubule damage can occur in the absence of glomerular damage [46,47]. It is possible that aldosterone is having a direct effect on the proximal tubule. Consistent with this hypothesis, prior publications have demonstrated expression (protein and mRNA) of MR in proximal tubules [48,49,50] and have shown that aldosterone regulates sodium transport in proximal tubule cells by a classical MR pathway [48]. Furthermore, in the hypertensive trans- genic TG (mRen2) (Ren2) 27 rats MR antagonism improved proximal tubule integrity by modifying the redox-sensitive mTOR/S6K1 pathway [51]. While we did not see a decrease in albuminuria with MR blockade in the current study, we previously demonstrated that MR blockade reduces L-NAME/ANG II induced proteinuria in male and female rats, and male mice [39,40,41]. It may be that the dose of MR blockade used in the current study was insufficient to block ALDO’s effect on albuminuria but was sufficient to reduce proximal tubular injury. Previously, we also showed that MR blockade reduces L-NAME/ANG II induced vascular injury in the kidney as well as vascular damage in the myocardium [40]. Consistent with these earlier studies and with the finding that Kim-1 predicts both renal and CV disease in humans, we demonstrated that the cardiac injury/stroke composite score was strongly correlated with urinary KIM-1 in the L-NAME/ANG II rodent model.

Our results coupled with those of others suggest that MR activation acts at several points in the onset and the progression of CKD, specifically at the level of proximal tubule, glomerulus, and vasculature. Consistent with this proposal, other investigators showed that MR activation induces glomerular podocyte injury, causing the disruption of the glomerular filtration barrier and proteinuria and MR blockade reduces podocyte damage and proteinuria [33,52,53]. In addition, MR blockade has a potent anti-inflammatory and antifibrotic properties [54,55,56]. These preclinical studies shed light on the mechanisms by which MR antagonism reduces renal disease and are relevant to results of recent large-scale studies in patients with CKD and type 2 diabetes. This demonstrates that treatment with finerenone, a MR blockade, as compared with placebo reduces CKD progression by (hazard ratio, 0.82; 95% confidence interval [CI], 0.73 to 0.93; *p* = 0.001) and cardiovascular events by (hazard ratio, 0.86; 95% CI, 0.75 to 0.99; *p* = 0.03) [34].

Due to our prior published results that use of a lipophilic statin, such as simvastatin, leads to lower aldosterone levels [43], we had originally hypothesized that if simvastatin prevented damage, it would be secondary to reduction in aldosterone levels. However, our findings that simvastatin reduces urinary KIM-1 and KIM-1 immunostaining in proximal tubules did not appear to be mediated by decreases in blood pressure or aldosterone levels. There are at least three possible explanations for this apparent discrepancy. First, simvastatin could have a direct protective effect on proximal tubule damage. We could find no published data to support this possibility. Second, since our previous study reported a decrease in aldosterone production by zona glomerulosa cells with simvastatin in male rats [43], there could be a biological sex difference in the relationship between simvastatin and aldosterone levels. However, two publications from our group showed no interaction between statin use and aldosterone levels with sex in humans [43,57]. Third, simvastatin could have an effect not only on reducing aldosterone production but also on MR activity. There are no published reports of this possibility. Finally, our assessments are technically limited. It is possible that small decreases in aldosterone sufficient to modify KIM-1 levels did occur, but we were unable to detect these differences. In addition, it is possible that the stimulatory effects of a chronic ANG II infusion—a continuous ANG II infusion for three days longer than the duration of ANG II infusion in the previous human and animal studies [43,57]—may have partially overcome the inhibitory effects of simvastatin on aldosterone in the current study.

Our results are consistent with a study showing that a single oral dose of simvastatin prior to the induction of renal ischemia reduces tubulointerstitial KIM-1 immunoreactivity along with reducing impairments in peritubular microvascular permeability and perfusion, glomerular barrier disruption, tubular dysfunction, and acute kidney injury [58]. Furthermore, our results are consistent with preclinical studies showing that simvastatin decreases the progression of renal fibrosis [59], renal injury in ischemia-reperfusion [60]. Another lipophilic statin, fluvastatin, has been shown to reduce tubulointerstitial and podocyte damage in puromycin amino nucleoside induced nephrosis [61]. In our study, the beneficial effects of simvastatin on KIM-1 were not observed with pravastatin, suggesting that the effects of lipophilic and hydrophilic statins on proximal tubular damage may differ.

Consistent with these preclinical studies, statins were reported to reduce the risks of 1-year and in-hospital mortality in patients with dialysis-requiring acute kidney injury and to lower incidence of acute kidney injury in critically ill patients [62]. A large-scale clinical study in humans, Study of Renal and Heart Protection (SHARP), showed that simvastatin plus ezetimibe daily safely reduced the incidence of major atherosclerotic events in a wide range of patients with advanced chronic kidney disease [63]. Consistent with our finding that simvastatin but not pravastatin reduces KIM-1, lipophilic statins (atorvastatin) were reported to be more protective than hydrophilic statins (rosuvastatin) in chronic kidney disease [64,65]. The mechanisms by which simvastatin but not pravastatin reduces KIM-1 are unknown. It is possible that the mechanisms are related to the differences in lipophilicity.

## 4. Methods

### 4.1. Study Approval

This study protocol was reviewed and approved by of the Institutional Animal Care and Use Committee at Brigham and Women’s Hospital, approval number is 2016N000387. All experimental procedures were conducted in accordance with the National Institutes of Health Guide for the Care and Use of Laboratory Animals and the guidelines of the Institutional Animal Care and Use Committee at Brigham and Women’s Hospital.

### 4.2. Animals

The 8–10-week-old female Wistar rats (Charles River Lab, Wilmington, MA, USA), weighing 180 to 260 g were purchased. The rats were housed in the animal facility in a 12-h light/dark cycle at 22 ± 1 °C ambient temperature. All animals were allowed one week to acclimate after arrival and unlimited access to rodent chow (Purina, St. Louis, MO) and water until the initiation of the experiment.

### 4.3. Experimental Study Protocol

The experimental study for each cohort was initiated at day 1 and completed at day 15 (Figure 1). Three days prior to the study initiation, all rats were housed in individual cages and started to receive a high salt diet (1.6% High Na^+^ 1.1% Potassium (Purina), Scott Pharma, Marlborough, MA, USA, CAT # 26661). They continued to receive high salt diet until the end of the study. One day prior to study initiation, rats were randomized to the following conditions for 14 days: control group received only high salt; L-NAME/ANG II treated group received L-NAME (0.2 mg/mL in drinking water) and ANG II (225 ug/kg/day for days 12–14 only); L-NAME/ANG II/eplerenone treated group received L-NAME (0.2 mg/mL in drinking water), ANG II (225 ug/kg/day for days 12–14 only) and eplerenone (100 mg/kg/day p.o.); L-NAME/ANG II/pravastatin treated group received L-NAME (0.2 mg/mL in drinking water), ANG II (225 ug/kg/day for days 12–14 only) and pravastatin (hydrophilic statin-20 mg/kg/day p.o.); L-NAME/ANG II/simvastatin group received L-NAME (0.2 mg/mL in drinking water), ANG II (225 ug/kg/day for days 12–14 only) and simvastatin (lipophilic statin-20 mg/kg/day p.o.). L-NAME was purchased from Sigma (Burlington, MA). The concentration of L-NAME in the drinking water was set as 0.2 mg/mL in drinking water to achieve a dose of approximately 40 mg/kg/day. Eplerenone (AstaTech, Bristol, PA, USA, CAT# 34998), pravastatin (Selleckchem, Houston, TX, USA, CAT# S3036) and simvastatin (AstaTech, Bristol, PA, USA, CAT#35892) treatments were given in rodent chow diet. On day 12, ANG II was administered via osmotic minipumps (Alzet minipump—model 2001, Alza Corp., Palo Alto, CA, USA), which were implanted in each rat subcutaneously under isoflurane anesthesia. ANG II (human, 99% peptide purity, A9525-5 × 1 mg) was purchased from Sigma (Burlington, MA, USA). Concentrations of ANG II to fill the minipumps were calculated based on the mean pump rate provided by the manufacturer and the body weight of the animals on the day prior to implantation of the pumps. The time of sacrifice for all study animals was on day 15 between 8.30 and 11 am except for the two rats who demonstrated evidence of a stroke on day 14 and were sacrificed on day 14. In total, four of fifty rats demonstrated evidence of a stroke (inability to keep balance and to move extremities on one side of the body), two on day 14 and two on day 15. The two rats which demonstrated evidence of a stroke on day 14 are defined as the animals with *early stroke*.

### 4.4. Blood Pressure Measurements

Systolic and diastolic blood pressure (SBP; DBP) were assessed by tail-cuff plethysmography (CODA High Throughput System, Noninvasive Blood Pressure System, CODA-HT8; Kent Scientific Corp., Torrington, Connecticut). SBP and DBP were measured in restrained, conscious rats as previously reported by our group [66,67]. Restrained rats were warmed to 35 °C on the warming plate and allowed to rest quietly in the room and then blood pressure measurements were performed by the same person throughout the study. The rats were not restrained longer than 15 min; this is the approach our group has been using for past 15 years [66,67]. Blood pressure measurements were performed early in the mornings between 8 and 10 AM after 3 rounds of acclimation over a one-week period. Values from 25 cycles were used to calculate the mean SBP and DBP for each cycle, as well as standard deviation (SD) for each rat at each time point. Any readings greater than two SDs from the mean were excluded. The final mean value of SBP and DBP for each group was calculated from average SBPs and DBPs of individual rats and presented as mean ± standard error of mean (SEM).

### 4.5. Tissue Collection

All rats were euthanized under anesthesia with isoflurane. Blood samples were collected via heart puncture and then hearts, adrenals, and kidneys were immediately excised. The upper one third of the heart and half of the right kidney (coronal section) were placed in cassettes and then kept in 10% formalin overnight, and then transferred to 70% ethanol, followed by processing by Harvard Medical Area Core Management System/BWH for histopathological processing (H&E staining for heart, PAS staining for kidney, KIM-1 immunostaining for kidney). The remaining parts of the heart and renal cortex were placed immediately in liquid nitrogen and then stored at −80 °C for further molecular analysis. Adrenals were placed on ice and dissected from surrounding fat tissue. The outermost capsulated portion, containing predominately zona glomerulosa cells, was carefully peeled away from the rest of the adrenal, and then stored at −80 °C for further molecular analysis.

### 4.6. Hormone Measurements

Blood samples were collected in BD Microtainer tubes with ethylenediamine tetra acetic acid (EDTA). Plasma was separated by centrifugation for 10 min at 1300 relative centrifugal force (RCF). We used the enzyme-linked immunosorbent assays (ELISA) aldosterone kit (IBL International GMBH, Hamburg, Germany) to measure plasma aldosterone levels. Plasma renin activity (PRA) levels were measured by the ELISA kit (IBL International GMBH, Hamburg, Germany), according to the manufacturer’s instructions. Plasma corticosterone levels were measured by the ELISA kit (Enzo Life Sciences, Inc., NY, USA).

### 4.7. Urine Analysis

Rats were placed in metabolic cages for four days from day 11 through day 15 with access to food and water; 24-h urine was collected each day. Urinary aldosterone levels were measured by ELISA assay (IBL International GMBH, Hamburg, Germany). Urinary albumin levels were measured using the antibody-based test, rat-specific ELISA assay (Alpco Diagnostics—Salem, NH) and urinary creatinine was measured by Quest diagnostics (1967, NJ). The 24-h urinary creatinine was used as an indicator of urine collection accuracy. In brief, average daily 24-h urine creatinine (mg/24-h) was determined for each animal based on five day 24-h urine collections. If the 24-h urinary creatinine for a specific day deviated by more than 15% from the average 24-h urine creatinine for that animal, then the average daily 24-h urine creatinine was recalculated excluding the out-of-range urine. A correction factor (the ratio of recalculated average daily creatinine to daily creatinine ratio) was applied if the urinary creatinine/24 h for a specific day deviated by more than 15% from that animal’s average 24-h urinary creatinine. All urine measurements were reported after application of the correction factor. Correction factors were applied for 14% of urines.

Kidney injury molecule-1 (KIM-1), proximal tubular injury biomarker levels were measured in 24 h-urine using an ELISA assay kit (Cat#RKM 100, RKM100, R&D Systems, Minneapolis, MN, USA) in pg/mL, according to the manufacturer’s instructions. The 24-h urinary KIM-1 levels were reported as KIM-1/Creatinine ratios in ng/mg.

### 4.8. Heart and Kidney Histopathologic Evaluation

Cardiac tissue, including both ventricles, was fixed and embedded in paraffin blocks for histopathologic analysis. Tissue was stained with hematoxylin and eosin and Masson trichrome for light microscopic analysis. Histologic evaluation was based on previous work by our group (Rocha et al., 2000). The two sections of the heart for each rat were evaluated by two pathologists blinded to treatment, and the reviewers’ scores were averaged for each rat. Myocardial injury, as defined by myocyte eosinophilia and nuclear drop-out, or frank necrosis, was expressed numerically as a percentage of involved myocardium on a scale from 0% to >30%. A cardiac injury score (CIS) of 0 was given for sections with less than 1% damage, CIS of 1 for damage more than 1% and equal or less than 8%, CIS of 2 for damage more than 8% and equal or less than 15%, CIS of 3 for damage more than 15% and equal or less than 22%, CIS of 4 for damage more than 22% and equal or less than 29%, and CIS of 5 for damage involving more than 30% of the myocardium.

We developed a cardiac injury/stroke composite score to provide a semi-quantitative assessment encompassing cardiac injury score and stroke status. The animals that exhibited evidence of stroke received a score of 6 if symptoms were present on day 14, and a score of 5 if symptoms were present on day 15. The cardiac injury/stroke composite score for each animal was equal to the maximum score of either stroke score or CIS score.

Kidneys were bisected, and a single cross section fixed and embedded in paraffin blocks. Each section was stained with KIM-1 antibody (TIM-1/KIM-1/HAVCR Antigen Affinity-purified Polyclonal Antibody, BioTechne Corporation, Catalog # AF3689) and developed with the Anti-Goat HRP-DAB Cell & Tissue Staining Kit (brown, BioTechne Corporation, Catalog # CTS008). One section of kidney was evaluated by two pathologists blinded to therapy, and reviewers’ scores were averaged for each rat. The percentage of renal tubules showing KIM-1 expression in the cortex and KIM-1 staining intensity was scored semi-quantitatively by each reviewer. Score 0 indicates no KIM-1 staining in the tubules, score 1 indicates 1–20% KIM-1 staining in the tubules with weak to moderate staining, score 2 indicates ≤20% KIM-1 staining in the tubules with strong staining or 30–50% KIM-1staining in the tubules with weak to moderate staining, and score 3 indicates ≥50% involvements with strong staining or ≥60% involvement with any staining.

### 4.9. mRNA Expression Analysis

Total RNA was extracted from kidney, heart and adrenal tissue using the RNeasy mini kit (Qiagen, Germantown, Maryland). cDNA was synthesized from 1.5 μg RNA with the first-strand cDNA synthesis kit (GE Healthcare, Piscataway, NJ, USA). Real-time Polymerase Chain Reaction (PCR) amplification reactions to detect Kim1, CYP11B2, and the housekeeping 18S ribosomal RNA were performed in duplicate using TaqMan gene expression assays (proprietary primers and probes designed and synthesized by Applied Biosystems, Foster City, California) using the QuantStudio 3 (Applied Biosystems). The mRNA expression levels were normalized to 18S ribosomal RNA levels and were determined using the ΔΔCT method. The data were presented as a fold increase relative to the control group.

### 4.10. Statistical Analyses

Data were analyzed by JMP Pro 15. Normality was verified by the Shapiro–Wilk test. Continuous variables were presented either as the medians ±95% confidence interval for non-normally distributed data, and as the means ± SEM (standard error of the mean) for normally distributed data. All p values were calculated 2-sided, and values less than 0.05, were considered statistically significant. For normally distributed data, the difference between the means of two groups were tested by t test, and the difference between the means of independent (unrelated) groups (i.e.; five treatment groups) at one time point were tested with one-way ANOVA. If the ANOVA indicated significant differences, we performed comparison for all pairs using the Tukey–Kramer HSD. In addition, to analyze data across time within a single treatment group, we used a mixed-model repeated-measures analysis. Statistical analyses of continuous variables which did not hold the normal distribution assumptions were performed by nonparametric tests; Wilcoxon/Kruskal Wallis tests (rank-sums). If the Wilcoxon/Kruskal Wallis tests (rank-sums) indicated a significant difference, we performed nonparametric comparisons for each pair using Wilcoxon method. Correlation analysis of these numeric continuous variables was performed using Spearman rank correlations.

## 5. Conclusions

The mineralocorticoid receptor blockade eplerenone reduced L-NAME/ANG II induced increases in KIM-1, a marker of proximal tubule damage, as well as a composite score of cardiac injury and stroke. Further, the lipophilic statin simvastatin, but not hydrophilic statin pravastatin, reduced L-NAME/ANG II induced increases in KIM-1, raising the possibility that reducing proximal tubular damage is another pleiotropic effect of simvastatin.

### 5.1. Contribution to Clinical Practice

Our findings suggest that both MR blockade and simvastatin reduce KIM-1, a marker of proximal tubular damage, and a strong predictor of CKD. Thus, decreases in proximal tubular damage may be an important mechanism by which MR blockade and simvastatin reduce CKD progression. Finally, KIM-1 may be a potential biomarker for excess MR activity.

### 5.2. Limitations of the Study

A limitation of our study is that we studied only female rats. However, we would anticipate similar results in male rodents, but this is yet to be proven. Since both MR blockade and simvastatin reduced KIM-1, future studies are needed to determine whether their effects would be additive.

## Figures and Tables

**Figure 1 ijms-24-06500-f001:**
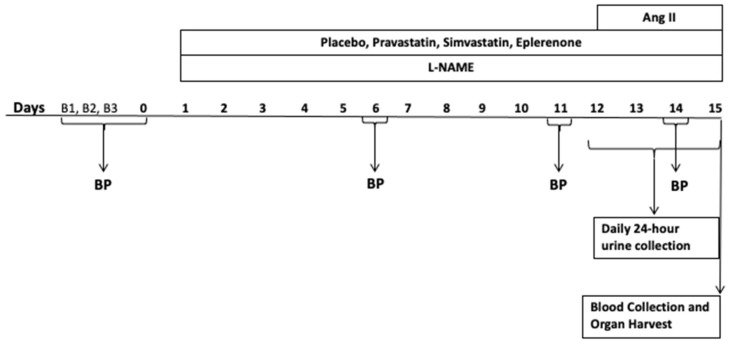
Experimental Study Timeline. All the rats received food containing 1.6% Na^+^ from day (−2) through day 14. The rats received Nω-Nitro-L-arginine methyl ester hydrochloride (L-NAME) in drinking water from day 1 through day 14 and ANG II infusion from day 12 through day 14 via osmotic minipumps. Blood pressures were measured noninvasively via tail cuffs three times at baseline (B1, B2 and B3) and on days 6, 11, and 14 during the study. The 24-h urine collections were performed from day 11 through day 14. Blood collection and organ harvest (hearts, kidneys, adrenals, and aortas) were performed on day 15.

**Figure 2 ijms-24-06500-f002:**
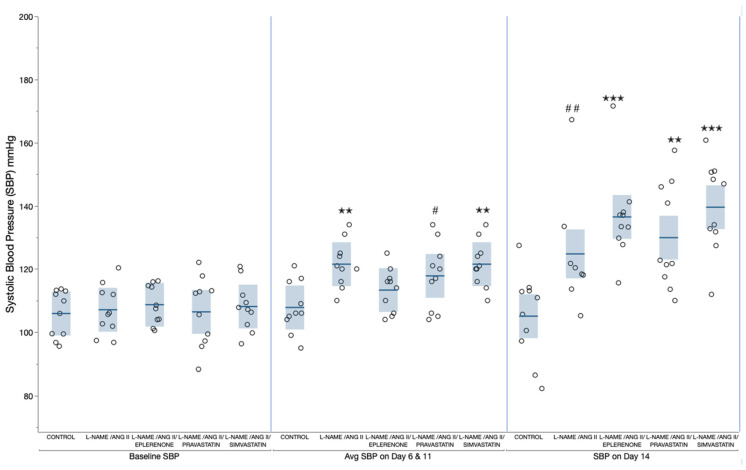
Each circle represents systolic blood pressure (SBP) for an animal. Data are presented as mean ± SEM. Data were analyzed by one-way ANOVA with subsequent pair wise comparison (Tukey–Kramer HSD) between treatment groups. Each of the following time points were analyzed separately; baseline blood pressure, average of day 6 and 11 blood pressure, and day 14 blood pressure. *p* values for comparison with the control group (Tukey–Kramer HSD) are as follows: ^✭✭^
*p* < 0.01; ^✭✭✭^
*p* < 0.001; ^#^
*p* = 0.06, ^##^
*p* = 0.07.

**Figure 3 ijms-24-06500-f003:**
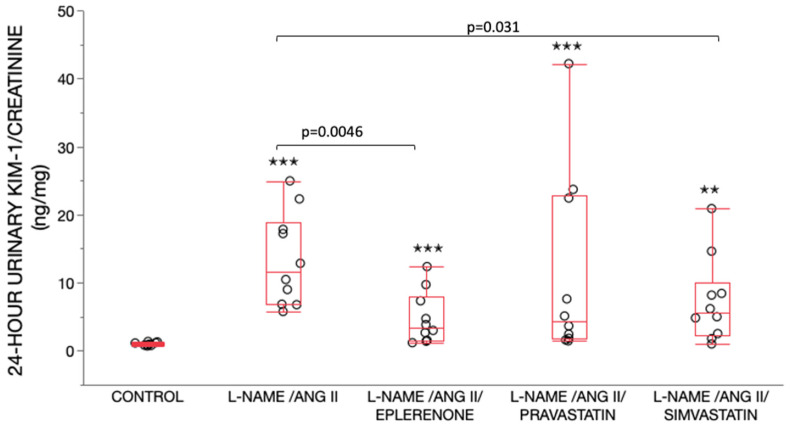
Day 14, 24-h urinary KIM-1/creatinine ratios (ng/mg). Data are shown as box plots (minimum value, first quartile, median, third quartile, and maximum value). The lines are superimposed in the control group. There were significant differences between the five treatment groups (*p* < 0.0001) by Wilcoxon/Kruskal Wallis (Rank sums) test. Nonparametric pairwise comparisons were performed using Wilcoxon method. There was a significant increase in day 14 urinary KIM-1/creatinine ratio (ng/mg) in all L-NAME/ANG II groups as compared with the control group (^✭✭^
*p* < 0.01, ^✭✭✭^
*p* < 0.001). Day 14 urinary KIM-1/creatinine ratio (ng/mg) was significantly lower in L-NAME/ANG II groups receiving eplerenone (*p* = 0.0046) and simvastatin (*p* = 0.031) as compared with the L-NAME/ANG II group.

**Figure 4 ijms-24-06500-f004:**
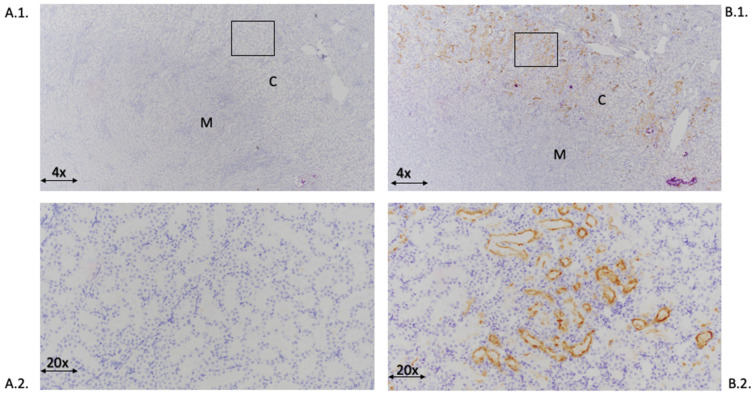
Immunostaining of KIM-1 in kidneys from a control treated animal (**A.1.**,**A.2.**) and L-NAME/ANG II treated animal (**B.1.**,**B.2.**). (**A.1.**) shows minimal to no KIM-1 immunostaining. B.1 shows KIM-1 immunostaining of proximal renal tubules. Boxes indicated the area in (**A.1.**) magnified in (**A.2.**) and the area in (**B.1.**) magnified in (**B.2.**). (**A.1.**,**B.1.**) 4× magnification, (**A.2.**,**B.2.**) 20× magnification. C indicates the renal cortex and M indicates the renal medulla.

**Figure 5 ijms-24-06500-f005:**
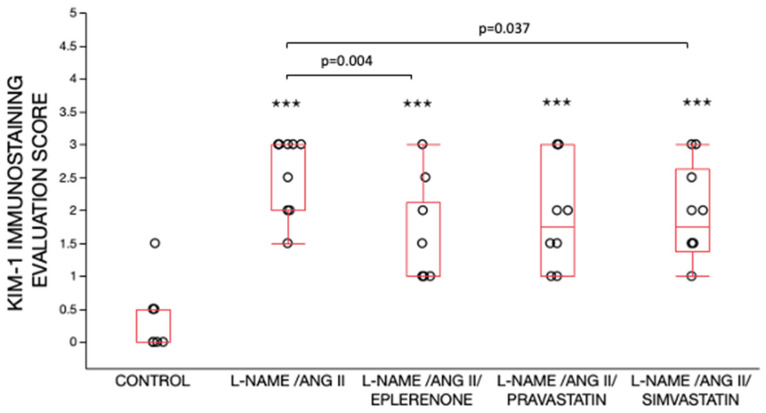
Semi-quantitative evaluation of kidneys with KIM-1 immunostaining. Data are shown as box plots. There were significant differences between the five treatment groups (*p* < 0.0001) by Wilcoxon/Kruskal Wallis (Rank sums) test. Nonparametric pairwise comparisons were then performed using Wilcoxon method. Higher KIM-1 immunostaining scores were observed in all L-NAME/ANG II groups as compared to the control group, ^✭✭✭^
*p* < 0.001. KIM-1 immunostaining scores were significantly lower in the L-NAME/ANG II/eplerenone (*p* = 0.004) and L-NAME/ANG II/simvastatin (*p* = 0.037) groups as compared with the L-NAME/ANG II group.

**Figure 6 ijms-24-06500-f006:**
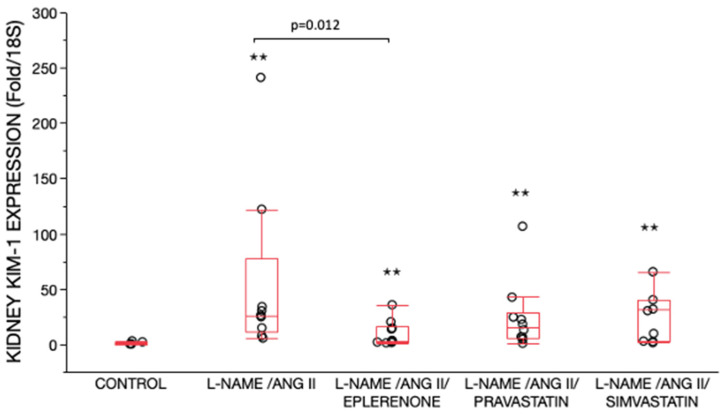
KIM-1 expression relative to expression in control group’s kidney. Data are shown as box plots (minimum value, first quartile, median, third quartile, and maximum value). The lines are superimposed in the control group. There were significant differences between the five treatment groups (*p* < 0.0006) by Wilcoxon/Kruskal Wallis (Rank sums) test. Nonparametric pairwise comparisons were then performed using Wilcoxon method. All L-NAME/ANG II groups had higher KIM-1 mRNA expression in renal cortex as compared with the control group, ^✭✭^
*p* < 0.01. KIM-1 mRNA expression was significantly lower in the L-NAME/ANG II/eplerenone (*p* = 0.012) group as compared with the L-NAME/ANG II group.

**Figure 7 ijms-24-06500-f007:**
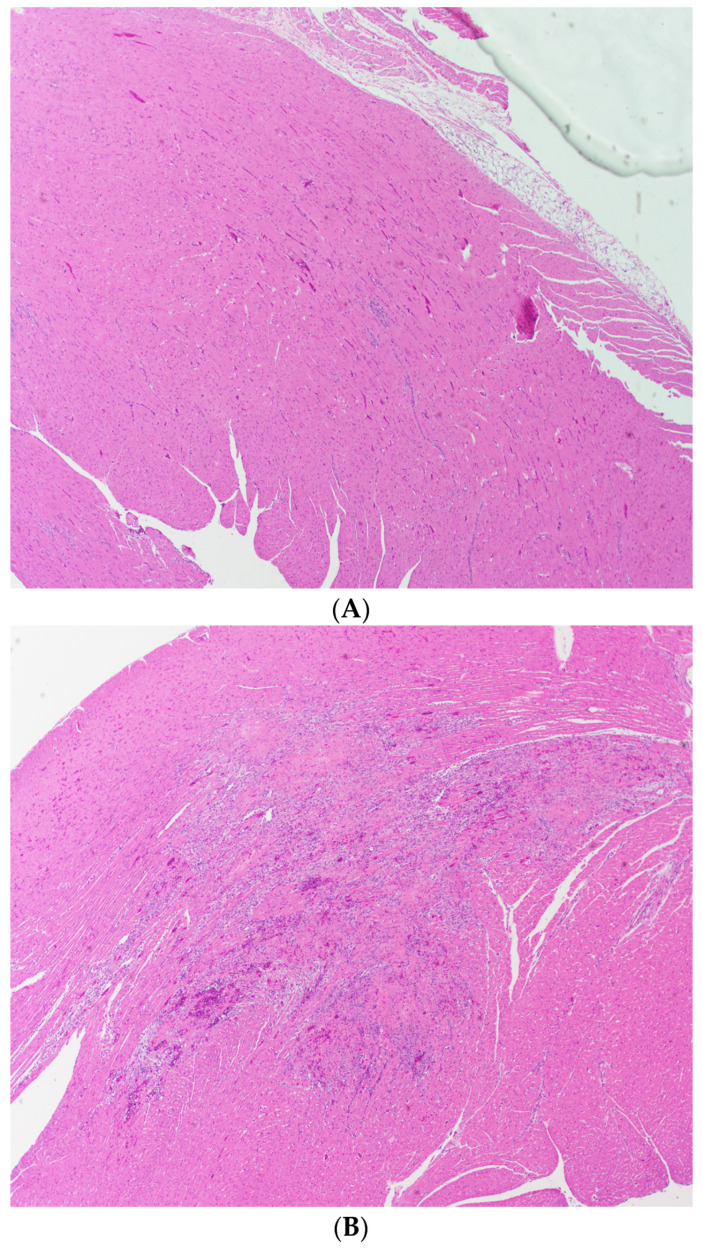
Cardiac Ventricle Histopathology. Representative section of cardiac ventricle stained with H&E from: (**A**) control group showing normal cardiac histology and (**B**) L-NAME/ANG II group showing prototypical cardiac damage characterized by myocyte eosinophilia, nuclear drop-out, myocyte necrosis, and inflammatory infiltrates; 4× magnification.

**Figure 8 ijms-24-06500-f008:**
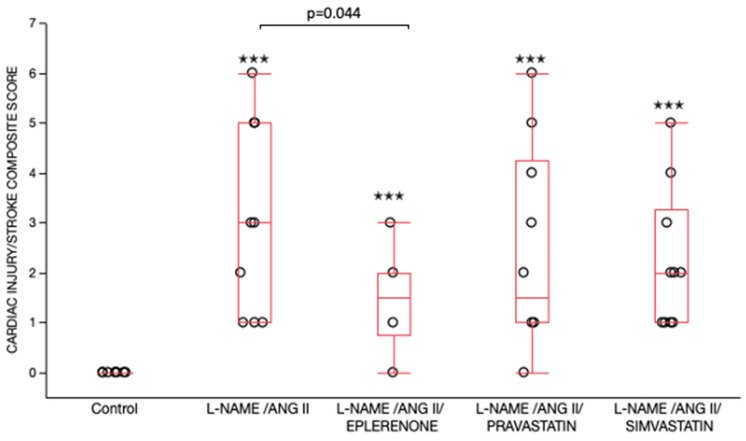
Cardiac Injury/Stroke Composite Score Comparison. Data are shown as box plots (minimum value, first quartile, median, third quartile, and maximum value). The lines are superimposed in the control group. There were significant differences between the five treatment groups (*p* < 0.001) by Wilcoxon/Kruskal Wallis (Rank sums) test. Nonparametric pairwise comparisons were performed using Wilcoxon method. All L-NAME/ANG II groups had significantly higher cardiac injury/stroke composite scores as compared to the control group, ^✭✭✭^
*p* < 0.001. The L-NAME/ANG II/eplerenone group (*p* = 0.044) had a significantly lower cardiac injury/stroke composite score compared with the L-NAME/ANG II group.

**Table 1 ijms-24-06500-t001:** Body weight and organ weight for the five treatment groups.

**Treatment Groups**	**Control**	**L-NAME/ANGII**	**L-NAME/ANGII** **Plus** **Eplerenone**	**L-NAME/ANGII** **Plus** **Pravastatin**	**L-NAME/ANGII** **Plus** **Simvastatin**	***p* Value**
**Body Weight (g)**						
**Baseline**	225.5 ± 2.7	226.2 ± 4.8	230.7 ± 4.3	230 ± 4.1	231.1 ± 5	0.81
**Day 11, 12** **(average)**	263.5 ± 4.7	262.6 ± 4.5	249.4 ± 2.03	257.5 ± 4.5	250.9 ± 7.42	0.16
**Day 15**	264.1 ± 6.2	239.9 ± 3.8 ^✭^	235.8 ± 5.3 ^✭^	241 ± 4.5 ^#^	233.1 ± 8.3 ^✭✭^	0.0039
**Organ Weight (g)**						
**Heart**	0.9 ± 0.03	1 ± 0.02	0.9 ± 0.01	0.9 ± 0.02	0.9 ± 0.03	0.25
**Heart/BW**	0.35 ± 0.01	0.37 ± 0.01	0.36 ± 0.01	0.35 ± 0.01	0.35	0.39
**Kidney**	1 ± 0.05	0.9 ± 0.04	0.9 ± 0.03	1 ± 0.03	0.9 ± 0.05	0.28
**Kidney/BW**	0.37 ± 0.02	0.34 ± 0.01	0.35 ± 0.01	0.37 ± 0.01	0.36 ± 0.01	0.25

Values represent means ± SEM. The last column displays the *p* value for one-way ANOVA between the five treatment groups. *p* values for comparison with the control group (Tukey–Kramer HSD) are as follows: ^✭^
*p* < 0.05; ^✭✭^
*p* < 0.01; ^#^
*p* = 0.06.

**Table 2 ijms-24-06500-t002:** Plasma and Urine Hormone Measurements, Albuminuria and CYP11B2 expression.

Treatment Groups	Control	L-NAME/ANGII	L-NAME/ANGII Plus Eplerenone	L-NAME/ANGII Plus Pravastatin	L-NAME/ANGII Plus Simvastatin	*p* Value
Plasma Collected at Day 15						
**Plasma** **Renin Activity** **(ng/mL·h)**	12.5 (−1.1–39.5)	2.7 ^✭^ (−1.7–14)	3.6 (−1.2–17.9)	1.53 ^✭✭^ (0.5–4.7)	1.5 ^✭^ (0.4–6.3)	0.047
**Aldosterone** **(ng/dL)**	13.7 (10.3–24.3)	77.7 ^✭✭✭^ (56.5–251.4)	149.5 ^✭✭^ (62.7–202.9)	73.7 ^✭✭^ (47.4–259.3)	67.3 ^✭✭✭^ (34.9–143.4)	0.0008
**Corticosterone** **(ug/dL)**	2.18 (1.1–4)	8.5 ^✭✭^ (4.9–14.3)	1.5 ^†^ (0.5–6.7)	7.4 ^✭^ (4–14.5)	2.9 ^†^ (0.8–9.1)	0.0053
Urinary Aldosterone (ng/24 h)						
**Day 11**	65.6 ^δδ^ (58.1–77.1)	63 ^δδ^ (51.2–74.1)	132.4 (98.3–172.7)	58 ^δδδ^ (48.5–71.8)	53.4 ^δδδ^ (49.4–68.4)	0.0005
**Average of** **Day 12–13–14**	65.4 (60.1–74.9)	258.6 ^✭✭✭^ (98.5–373.6)	310.5 ^✭✭✭§^ (208.9–506.2)	179.2 ^✭^ (103.1–293.8)	128.4 ^✭✭✭^ (96.4–213.9)	<0.0001
Albumin to Creatinine Ratio (mg/mg)						
**Day 14**	0.04 (0.03–0.04)	2.1 ^✭✭✭^ (1–4.6)	1.4 ^✭✭✭^ (0.2–4.7)	4.5 ^✭✭✭^ (1.7–10)	3.3 ^✭✭✭^ (1.6–7)	<0.0001
**Adrenal CYP11B2**						
**CYP11B2 mRNA (relative expression)**	0.79 (0.27–1.73)	212.1 **^✭✭^** (107.1–1000)	425.2 **^✭✭✭^** (261.6–1072.5)	130.8 **^✭✭✭^** (36–455)	227.9 **^✭✭^** (79.9–549.5)	**0.0004**

Values represent the medians and 95% CI. Differences between the five treatment groups were analyzed by Wilcoxon/Kruskal Wallis (Rank sums) tests (*p* value displayed in the last column), followed by nonparametric pairwise comparisons using Wilcoxon method; ^✭^
*p* < 0.05, ^✭✭^
*p* < 0.01, ^✭✭✭^
*p* < 0.001 for comparison with control group; ^†^
*p* < 0.05 for comparison with the L-NAME/ANG II group; ^δδ^
*p* < 0.01, ^δδδ^
*p* < 0.001 for comparison with the L-NAME/ANG II/eplerenone group; ^§^
*p* < 0.05 for comparison with the L-NAME/ANG II/simvastatin group.

## Data Availability

The data that support the findings of this study are openly available in a data repository website, “figshare” at https://doi.org/10.6084/m9.figshare.21385710, accessed on 23 October 2022.

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
