# Peer review of "Effects of Mineralocorticoid Receptor Blockade and Statins on Kidney Injury Marker 1 (KIM-1) in Female Rats Receiving L-NAME and Angiotensin II"

_ijms, 2023, doi:10.3390/ijms24076500_

Round 1

Reviewer 1 Report

In this manuscript, Huang et. al. explore the effects of simvastatin and eplerenone treatment on renal damage on rats given L-NAME and AgnII. Based on robust KIM-1 measurements, the authors conclude that MR blockage prevents proximal tubule damage in L-NAME/AngII treated rats, independently of blood pressure or albuminuria.  Authors argue that the dissociation between glomerular, and proximal tubule damage may be due to an incomplete effect of eplerenone on blocking MR. Simvastatin treatment yields similar results, even though it failed to significantly reduce aldosterone levels in plasma of female rats. The manuscript is well written and data clearly presented for the most part of it. Some comments follow.

Minor comments:

·        The term “liberal salt” is ambiguous. Specific terms as “High salt diet” or “elevated Na intake” are preferred.

·        Figure 6 should be Figure 1, and referred after line 88 of the introduction.

·        Text lines restart in page 8.

·        Source of some drugs in missing in the methods section.

Major comments:

1.      Blood pressure data should be presented in a figure as opposed to a table. A chart better represents longitudinal data. In addition, an explanation should be provided for averaging days 6 and 11. The figure legend should clearly explain which groups were compared.

2.      Authors should provide an explanation on why the eplerenone and simvastatin groups have higher blood pressures that L-NAME/AngII.

3.      Authors should reconcile the fact the simvastatin inhibits aldosterone production with the fact that aldosterone levels did not change. This could be a sexual dimorphism, as authors suggest in the discussion. I would be convincing to see some data supporting this.

4.      In addition, the link between MR activation and PT damage is unclear, given that BP and albuminuria were not resolved or improved. Authors should offer an explanation on how they think these effects are occurring. Do proximal tubules express MR? Some publicly available single-cell transcriptional data may help clarify this issue.

Reviewer 2 Report

Huang et el using the rodent L-NAME/angiotensin II renal injury model investigated the effects of eplerenone (mineralocorticoid receptor agonist) and simvastatin on the expression of the renal injury biomarker KIM-1. The manuscript is well written and provides increment evidence that diminished mineralocorticoid receptor activation limits proximal tubular damage.

Comments:

1/ Abstract, line 15. “Aldosterone … causes cardiac and renal injury.” Moderate – suggest the word “causes” be replaced with “is linked to”

2/ Methods section, lines 267. Given the small sample size (n=10 per group) data should not be excluded on the basis that the values are 2.6 or more standard deviations of the mean. You state you are using nonparametric (rank based) statistical methods – and as such, the data will carry no more weight than being first ranked out of 10.

Round 2

Reviewer 1 Report

This is a second revision round of the manuscript by Huang et.al. Authors have answered previous concerns; however, issues with the interpretation of the data persist.

Major comments:

The scope of this investigation is “to determine the impact of modulating MR activity on PT injury as assessed by KIM-1”. This idea is supported by previous studies in which administration of AngII increased KIM-1. An assumption was made that such increase in KIM-1 was due to activation of MR, as opposed to direct effects of AngII. It was also suggested that the increase in KIM-1 in the DOCA-salt model was due to the elevated blood pressure, as spironolactone did not prevent it. Additionally, Author’s previous reports indicate the simvastatin blocks aldosterone production.

Given this background, the expected results where that in the L-NAME/AngII model, eplrerenone and pravastatin would prevent elevations in KIM-1. Eplerenone by reducing blood pressure, and simvastatin by reducing both, aldosterone and BP. However, that was not the case. Both eplerenone and simvastatin reduced KIM-1, but paradoxically, they increased blood pressure. In addition, simvastatin failed to reduce aldosterone in the L-NAME/AngII group as compared L-NAME/AngII without the statin.

In its current version, the manuscript lacks mechanistic data on how eplerenone or simvastatin prevents the increase in KIM-1, as well as a satisfactory explanation for the increase in blood pressure ion these groups.

Minor comments:

The statistical analysis conducted on each table/figure is unclear

Blood pressure should be presented as BP:F(time). All groups and time points in one graph.

Round 3

Reviewer 1 Report

Thanks for addressing my comments